# Relaxation dynamics in bio-colloidal cholesteric liquid crystals confined to cylindrical geometry

Sayyed Ahmad Khadem[1,2,5], Massimo Bagnani [3,5], Raffaele Mezzenga [3,4] & Alejandro D. Rey [1,2✉]

Para-nematic phases, induced by unwinding chiral helices, spontaneously relax to a chiral ground state through phase ordering dynamics that are of great interest and crucial for applications such as stimuli-responsive and biomimetic engineering. In this work, we characterize the cholesteric phase relaxation behaviors of β-lactoglobulin amyloid fibrils and cellulose nanocrystals confined into cylindrical capillaries, uncovering two different equilibration pathways. The integration of experimental measurements and theoretical predictions reveals the starkly distinct underlying mechanism behind the relaxation dynamics of β-lactoglobulin amyloid fibrils, characterized by slow equilibration achieved through consecutive sigmoidal-like steps, and of cellulose nanocrystals, characterized by fast equilibration obtained through smooth relaxation dynamics. Particularly, the specific relaxation behaviors are shown to emerge from the order parameter of the unwound cholesteric medium, which depends on chirality and elasticity. The experimental findings are supported by direct numerical simulations, allowing to establish hard-to-measure viscoelastic properties without applying magnetic or electric fields.

[1] Department of Chemical Engineering, McGill University, Montreal, QC H3A 2B2, Canada. [2] Quebec Centre for Advanced Materials, Canada (QCAM/CQMF), Montreal, QC H3A 2K6, Canada. [3] Department of Health Sciences and Technology, ETH Zurich, Schmelzbergstrasse 9, Zurich 8092, Switzerland. [4] Department of Materials, ETH Zurich, Wolfgang-Pauli-Strasse 10, Zurich 8093, Switzerland. [5] These authors contributed equally: Sayyed Ahmad Khadem, Massimo Bagnani. ✉email: alejandro.rey@mcgill.ca

Unwinding the helices of a chiral nematic liquid crystal drives the cholesteric phase (N*) to a para-nematic phase (PN)[1], characterized by a higher energy state compared to the equilibrium chiral nematic phase. Thus, the para-nematic phase thermodynamically tends to relax by relieving the excess free energy[2]. Through this thermodynamically driven relaxation, unwound helices forming a para-nematic state are spontaneously self-reconstructed leading to a chiral ground state; the PN–N* transition is thus self-driven. The dynamics of self-reconstruction (relaxation) in biological chiral lyotropic liquid crystals (BCLLCs) is of particular interest for forming retarder films and cholesterol color filters[3], plasmonic mesostructured materials[4], stimuli-responsive materials design[2,5–8], and processing of advanced materials[9] such as biomimetic film formation[10,11] replicating, for example, structural motifs of cortical bone and cornea[12,13]. Despite its importance, the prediction and quantification of the relaxation dynamics of BCLLCs have remained up to date non-trivial and challenging both experimentally and theoretically.

The confinement geometry can potentially affect relaxation dynamics as it has a considerable impact on the relaxed configuration[14–22]. The confinement geometry considered herein is a micron-sized cylindrical capillary for which the ground states of chiral mesogens have been extensively studied[17,18,23]. Given that the capillary diameter in our study is greater than the pitch length (i.e. $\frac{D}{p_\infty} = 6$ and $\frac{D}{p_\infty} = 13$ for BLG and CNC, respectively), the ground state of fibers macroscopic orientation eventually ends up into a concentric configuration in which the chiral helices are aligned along the cylinder diameters[17,19,24]. This microstructure is of importance in biomimicry as it, for example, mimics the osteon architecture which is the essential part of compact bone[13].

Furthermore, viscoelastic properties, such as elastic constants and rotational viscosity, play significant roles in the relaxation dynamics of liquid crystals (LCs). Another importance of viscoelastic properties is their usages in LC rheology nematodynamics, and flow-processing of fibers and films[1,25–31] such as biomimetic material design through coating[10]. Yet, the estimation of these properties has been a long-lasting challenge in the area of mesogenic solutions[32,33]. In particular, these properties strongly depend on concentration regimes; dilute $(c < \ell^{-3})$, semidilute $(\ell^{-3} < c < d^{-1}\ell^{-2})$, and concentrated $(c > d^{-1}\ell^{-2})$ where $c$, $\ell$, and $d$ denote number density, length, and diameter of fibers, respectively[32]. The first two regimes have been partially understood whereas the last one, in which a liquid-crystalline phase emerges, has not been adequately addressed, particularly for BCLLCs[32–34]. The difficulty in the estimation of viscoelastic properties of BCLLCs stems from the fact that standard techniques for viscoelastic properties measurements are based on applying a magnetic or electrical field to reorient mesogenic constituents[35–39] and, in general, these techniques are not readily applicable for BCLLCs due to their low diamagnetic and dielectric susceptibilities[39,40]. In this case, the convenient alternative is flow alignment[1,41–43]. To destabilize and unwind the chiral helices, rather than employing external fields such as magnetic or electric field, we thus rely on the surface tension-driven flow-induced orientation[25] created during capillary filling.

In the present study, we execute a methodology that allows quantifying spatio-temporal relaxation of chiral biological LCs confined to cylindrical capillaries, which corresponds to the spontaneous PN–N* transition. In particular, we characterize the relaxation behavior of BLG and CNC. Surprisingly, we uncovered that these two similar bio-colloidal LCs relax through considerably different pathways. BLG slowly relaxes through consecutive steps; each of these steps corresponds to a temporary slow formation of cholesteric layers, followed by a rapid equilibration which forms a sequence of plateaus, yielding slow–fast mode. The second system investigated, CNC, relaxes faster and with a smoother and continuous behavior characterized by the absence of plateaus, yielding smooth mode. The section "Distinct relaxation dynamics" elaborates these findings along with other different relaxation behaviors in detail. Thereafter, in the section "Mechanisms of BLG and CNC relaxations", we characterize and explain the essential mechanisms behind the novel relaxation dynamics. In this regard, we use direct numerical simulations showing that we can quantitively predict these relaxation dynamics in excellent agreement with experimental observations. After establishing the relaxation mechanisms, we reveal the physics behind the explored mechanisms and generalize the physical rules governing the relaxation dynamics in the section "Understanding the physical origins of relaxation mechanisms". In this section, our results reveal that the relaxation dynamics of BCLLCs confined to cylindrical capillaries generically obey the slow–fast or the smooth behaviors governed by a delicate interplay of chirality, viscoelasticity, and confinement geometry. Lastly, in the section "Properties estimation and relaxation time controllers", we propose a systematic framework to estimate fundamental viscoelastic properties without applying magnetic or electric fields, which is of particular interest for the BCLLCs. In addition to validating the estimated properties acquired by the proposed framework, the physical factors controlling the relaxation time are discussed.

## Results

**Distinct relaxation dynamics.** Figure 1a–l shows the sequences of microscopy images analyzed by PolScope[44]. The analysis allows extrapolating the average of fibers' orientation over the thickness of capillary tube and the two-dimensional (2D) fibers' orientation is represented according to the colormap shown in Fig. 1f; for example, the pink color which is ubiquitous in the POM images indicates that fibers are aligned parallel to the central axis (y-axis) (Fig. 1a–l, o). Initially (right after filling capillaries), the fibrils located close to the center of the cylindrical capillary form a para-nematic phase with a director field parallel to the central axis of the cylinder due to flow-induced alignment during capillary filling (described by pink color). Shortly thereafter (within the first 20 min), dark zones emerge in the center of the capillaries and some cholesteric fingerprints are already visible near the walls.

During equilibration, chiral fronts form and propagate from the wall inwards and gradually replace the dark areas, showing how the system is progressively equilibrated by the self-reconstruction of unwound chiral helices (Fig. 1a–l and Supplementary Movies 1 and 2). The physical origin of the initial dark areas emerging from the para-nematic phase cannot be interpreted unequivocally since the absence of birefringence can correspond to different scenarios. In fact, the fibers' orientation in the dark areas (i.e. capillary middle) can represent three different situations: (1) fibers do not possess mesogenic correlation and the phase could be isotropic although the fibers' concentration is a constant equal to the cholesteric bulk (i.e. the upper binodal miscibility boundary); (2) fibers are aligned normal to the plane and thus the phase is nematic; (3) the cholesteric helices are aligned normal to the plane and the light path, thus the phase could be already cholesteric. Determining the fibers' configuration in the dark areas is addressed below.

To rationalize the observations, we define the dimensionless normalized relaxation progress: $R(t) = \frac{\bar{q}(t)}{q_\infty}$, where $\bar{q}(t) = \frac{2\pi}{\bar{p}(t)}$, $\bar{q}(t)$ and $\bar{p}(t)$ are the spatially averaged chiral wavevector and pitch length, respectively. This quantity, $R$, provides insight into the

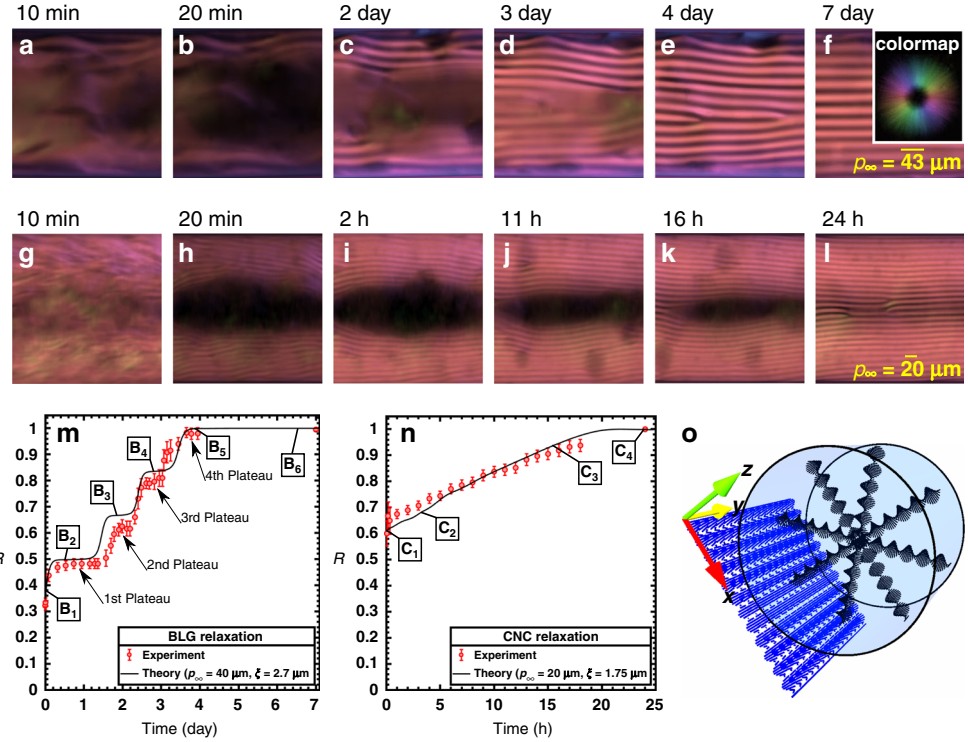

**Fig. 1 Slow–fast and smooth relaxation dynamics corresponding to BLG and CNC, respectively. a–f** Time-series snapshots of BLG microscopy (POM) images. **g–l** Time-series snapshots of CNC microscopy (POM) images. The images shown in panels **a–l** were experimentally acquired using the LC PolScope device and different colors represent different director field orientations which are appreciated according to the colormap depicted as the inset in the panel **f**. See Supplementary Note 1 for the detailed discussion on the mapping between fibers orientation and colormap. Note that the colormap shown as the inset in the panel **f** is applicable for all the microscopy images studied in this work. Over the relaxation time, cholesteric fingerprints progressively propagate into the unwound state (dark zone). Times elapsed from the beginning of relaxation are shown on top-left corners; min and h stand for minutes and hours, respectively. **m** Experimental observation and numerical simulation on the normalized relaxation progress curve, $R$, for BLG. The labels $B_1$ to $B_6$ correspond to Fig. 2. **n** Experimental observation and numerical simulation on the normalized relaxation progress curve, $R$, for CNC. The labels $C_1$ to $C_4$ correspond to Fig. 3. **o** The coordinate system used throughout this study along with a schematic illustration of a capillary tube indicating the ground state of fibers configuration in a circular cross-section in the $xz$-plane, known as concentric, and the averaged fibers configuration in a lateral plane in the $xy$-plane, known as chiral monodomain. The averaged fibers configuration on the lateral plane is representative of the colormap PolScop analysis shown in the panels **a–l**.

**Table 1 BLG and CNC properties obtained by the experiment and/or simulation.**

|  | $w_f$ (%)[a] | $\rho_f$ (g ml$^{-1}$)[a] | $.l$ (nm)[a] | $d$ (nm)[a] | $S_e$ | $\xi$ (μm) | $\eta$ (P) | $L_1$(N) |
|---|---|---|---|---|---|---|---|---|
| BLG | 2.8 | 1.3 | 652 ± 400 | 3.75 ± 0.8 | 0.66 | 2.70 | $1.1 \times 10^6$ | $7.8 \times 10^{-11}$ |
| CNC | 2.7 | 1.5 | 325 ± 168 | 4.5 ± 1.0 | 0.61 | 1.75 | $5.6 \times 10^5$ | $4.2 \times 10^{-11}$ |

[a]See Supplementary Note 2 for further information on the properties.

amount of space occupied by the cholesteric phase at any given time in the system. The relaxation progress, $R$, ranges from 0 to 1; for $R = 0$, there is no chiral nematic phase in the system, and when $R = 1$, the chiral nematic phase fills the sample and the system reaches the ground state. To obtain $\bar{q}(t)$, each time-series POM snapshot is partitioned into approximate monodomains, and $\bar{q}(t)$ is then computed as the weighted average of local chiral wavevectors: $\bar{q}(t) = \sum \frac{A_i q_i}{A}$ where $A$, $A_i$, and $q_i$ indicate the total area of the snapshot, area of the cholesteric zone, and chiral wavevector for the $i$th partition, respectively. Figure 1m, n shows the time evolution of $R$ for BLG and CNC, respectively. $R$ starts with an initial value representing the chiral nematic layers that are near the bounding surface at the very beginning of the experiment. Eventually, $R$ reaches unity, indicating the equilibration is complete all over the field of view and the concentric and monodomain configurations are formed in circular cross-sections

and lateral planes, respectively; schematically shown in Fig. 1o. Note that the monodomain fingerprint gradually appearing from top and bottom toward the center in the POM images is due to the spatial average of concentric configuration progressively forming from the capillary wall towards the capillary center. In other words, the 2D fingerprint on the lateral plane provides insight into the three-dimensional configuration exiting inside the capillary (see Fig. 1a–l, o).

BLG and CNC are similar from various viewpoints; these BCLLCs are aqueous solutions of semi-rigid rod-like bio-colloidal LCs characterized by a long pitch length ($p_{\infty,BLG} = 43$ μm and $p_{\infty,CNC} = 20$ μm), high aspect ratio ($\ell_{BLG} = 652$ nm and $\ell_{CNC} = 325$ nm, $D_{BLG} = 4$ nm, $D_{CNC} = 4.6$ nm, $(\ell/D)_{BLG} = 163$, and $(\ell/D)_{CNC} = 71$), similar polydispersity, critical concentrations, and densities (Supplementary Note 2, Table 1)[14,21,22,45,46]. Lastly, as estimated in this study, BLG and CNC possess similar

**Table 2 Order parameter and rotational viscosity of para-nematic mediums.**

|  |  | Phase (I) | Phase (II) |  | Equilibrium |
|---|---|---|---|---|---|
| BLG | $S_d$ | 0.33 | 0.23[a] | 0.33 | 0.66 |
|  | $\gamma(P) \times 10^{-5}$ | 8.73 | 9.87[a] | 8.73 | 3.5 |
| CNC | $S_d$ | 0 | 0 |  | 0.61 |
|  | $\gamma(P) \times 10^{-5}$ | 5.6 | 5.6 |  | 2.2 |

See Supplementary Note 5 and Movies 4 and 6. [a]Signifies the local minima.

rotational viscosity coefficients, $L_1$ Landau elastic constants, and coherence lengths (Tables 1 and 2). Surprisingly, although BLG and CNC have multiple physical properties that are similar and are confined into an identical capillary tube, $D = 260\,\mu m$, they manifest remarkably different relaxation according to these three criteria:

The first difference is the characteristic time required for the spontaneous PN–N* transition during which the fibers equilibrate from the unwound state (i.e. non-equilibrium para-nematic) to the chiral ground state, see Fig. 1. BLG closely reaches the equilibrium state 4 days after capillary filling. In the time span of 4–7 days, relaxation has minor progress and defects may be released over this period; see Fig. 1a–f, m. The relaxation of CNC suspension only takes 1 day to reach equilibrium; see Fig. 1g–l, n. Hence, the relaxation time of BLG is nearly four times longer compared to CNC. It is important to notice that at the beginning of the relaxation process, when compared with BLG, CNC has more intact chiral helices near the walls; see Fig. 1b, h. This fact is also highlighted in Fig. 1m, n, where $R$, evolves from 0.6 to 1 as relaxation progresses for CNC, while it goes from 0.3 to 1 for BLG. Indeed, the higher initial relaxation progress is, the shorter time would be expected for achieving the relaxed state; thus, the higher initial relaxation progress of CNC, $R = 0.6$, certainly plays role in shortening the relaxation time. To achieve a quantitative comparison between the two relaxation behaviors, we compute the relaxation rates for the two systems. The relaxation progress of CNC evolves by 40% over 1 day for reaching the equilibrium point ($R = 1$), while it evolves by 70% over 4 days for BLG and therefore the relaxation rates are 40% and 17.5% per day, respectively, confirming that CNC relaxation is more than two times faster compared to BLG.

The second difference between these two systems involves the number of trapped defects at equilibrium. As shown in Fig. 1a–l, the ground state of BLG shows fewer defects compared to CNC. It has already been shown that, for a given pitch length, more defects emerge provided the geometric size increases[47], here the geometric size is the diameter of the cylindrical capillary. Similarly, at a fixed capillary diameter $D = 260\,\mu m$, the number of defects increases when the pitch length is shorter. For BLG, $p_{\infty,\text{BLG}} = 43\,\mu m$, and for CNC, $p_{\infty,\text{CNC}} = 20\,\mu m$, corresponding to zero and four defects, respectively (see Fig. 1f, l). The ratio of geometric size to pitch length is thus an indicator for the number of generated defects; BLG relaxation gives rise to fewer defects compared to CNC suspension owing to $\frac{D}{p_{\infty,\text{BLG}}} = 6$ and $\frac{D}{p_{\infty,\text{CNC}}} = 13$. Another factor controlling the number of trapped defects is the relaxation time. Faster relaxation dynamics lead to more trapped defects as defects do not have time to be expelled. Thus, the CNC faster relaxation dynamic should be taken to account as a promoter of trapped defects. Taken together, the number of defects is directly and inversely proportional to $\frac{D}{p_{\infty}}$ and the relaxation time, respectively. Therefore, it is ideally expected that the defect-less ground state can be achieved as long as $\frac{D}{p_{\infty}}$ and the relaxation time are sufficiently small and long, respectively. It

should also be taken into consideration that the defect-less structure is achieved on a defined region of interest ($260\,\mu m \times 260\,\mu m$) for BLG as shown in Figs. 1a–f and 2; however, it is impractical to reach the defect-less structure on a large domain (e.g. $600\,\mu m \times 260\,\mu m$) even after equilibration, see Supplementary Note 3.

Although the defect analysis discussed above was performed on the region of interest ($260\,\mu m \times 260\,\mu m$), Supplementary Note 3 demonstrates that the chosen system size is statistically large enough to extend the concluded results regarding the number of trapped defects to larger regions. As shown in Supplementary Fig. 3, the number of defects trapped in the BLG system is less compared to CNC on a larger region ($600\,\mu m \times 260\,\mu m$) after equilibration (10 days for BLG and 3 days for CNC). Additionally, the defects existing in the capillary are mobile and thus can translate in and/or out the system investigated ($260\,\mu m \times 260\,\mu m$); however, the results already discussed are always statistically valid (see Supplementary Movies 1 and 2 and Supplementary Note 3).

Finally, these two materials differ on how the speed of relaxation changes over time. As shown in Fig. 1m, BLG relaxes over consecutive steps consisting of four plateaus, which we described as slow–fast relaxation behavior. During each plateau, there is no discernible change in the total area of the striped zone, reflecting that cholesteric phase formation slows down considerably, as discussed below. However, in the CNC suspension, the relaxation is characterized by a smooth behavior, where no clear plateau can be detected (see Fig. 1n).

The liquid-crystal rotational viscosity is the resistance to rotations of the average macroscopic director orientation[32,33]. For this reason, it is expected that the longer relaxation time observed for BLG can be attributed to higher rotational viscosity of BLG. However, there has not yet been any report on rotational viscosities and relaxation trends of these two bio-colloids, and the physical origins behind the different behaviors remain up to date unexplored. The solid lines shown in Fig. 1m, n are the predictions obtained by direct numerical simulations of the time-dependent **Q**-tensor continuum model of cholesteric liquid crystals that describes the total free-energy minimization. In our work, the net free energy of the system is comprised of free-energy functionals given by the well-established continuum theories of Landau-de Gennes (LdG) and Frank-Oseen-Mermin (FOM)[25,32,48,49] (see section "Direct numerical simulation"). As shown in Fig. 1m, n, the adopted theoretical approach predicts both the slow–fast and smooth relaxation dynamics for BLG and CNC, respectively. Hence, we rely on continuum liquid-crystal theory to first explore the physics behind the observations and secondly to estimate viscoelastic properties, including rotational viscosity coefficient which allows us rationalizing the fast and slow relaxation behaviors of CNC and BLG, respectively.

**Mechanisms of BLG and CNC relaxations.** As shown in Fig. 2, the theoretical and experimental results consistently show that cholesteric layers (i.e. the striped bands in POM images) are gradually formed from the top and bottom walls on the lateral plane. As can be seen in our results, the relaxation (the PN–N* transition) spontaneously takes place through the chiral front propagation; the cholesteric fingerprints radially propagate from the circular wall inward through the unwound para-nematic medium until the concentric configuration is achieved. This configuration is expected since the capillary diameter is larger than pitch length[17,19], $\frac{D}{p_{\infty}} = 6$ (Supplementary Note 4 and Movie 3).

The mechanism of BLG relaxation generally consists of two phases (Supplementary Movies 3 and 4) that can be described as

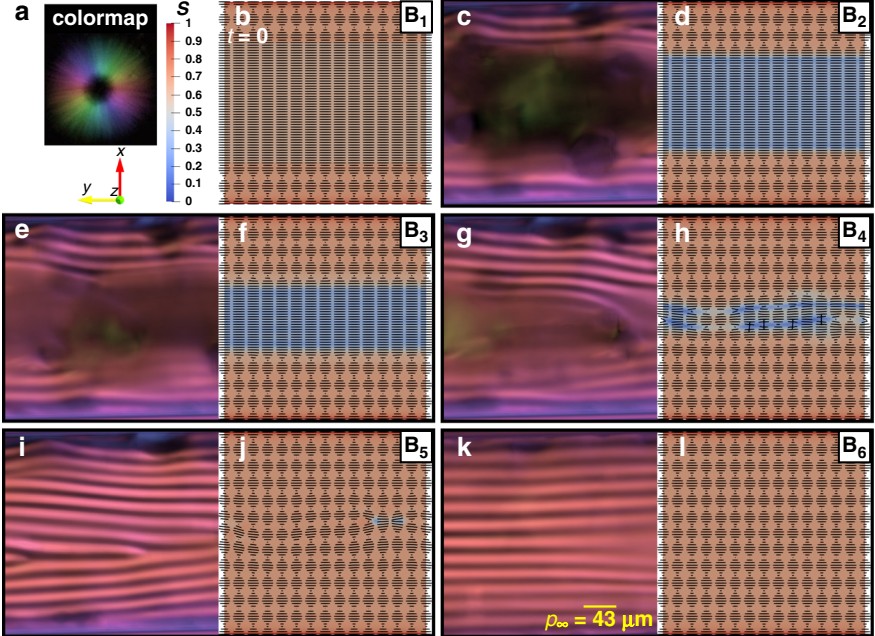

**Fig. 2 Mechanism of slow–fast dynamic in BLG relaxation. a** The colormap indicating the fibers orientation in the experimental POM panels captured by the LC PolScope device (see Supplementary Note 1 for further information), the blue-to-red spectrum showing the order parameter, $S$, computed by direct numerical simulation (see Supplementary Note 6 for further information), and the coordinate system. **b** Initialization of director field in direct numerical simulation, representing the initial configuration of fibers. **c**, **e**, **g**, **i**, **k** The experimental POM panels corresponding to the $B_2$ to $B_6$ stages shown in Fig. 1m. **d**, **f**, **h**, **j**, **l** The schematic fibers orientation and their order parameter computed by direct numerical simulation in the lateral plane for the $B_2$ to $B_6$ stages shown in Fig. 1m. General Note: the label numbers from $B_1$ through $B_6$ are marked in Fig. 1m.

*Phase (I)*: This phase takes place over a considerably short time span in the early relaxation. During Phase (I), all fibers, which are in the para-nematic state, lose their initial order parameter, $S_i = 0.6$ (ref. [42]), and then turn into a low-order-parameter para-nematic state (nearly isotropic), $S_{d,BLG} \approx 0.33$ (Fig. 2 and Supplementary Movies 3 and 4).

*Phase (II)*: Right after rapid formation of the para-nematic medium with a low-order parameter (i.e. Phase (I)), relaxation goes through Phase (II) in which a circular chiral front propagates and rewinds the fibers. As can be seen in Supplementary Movies 3 and 4, Phase (II) in BLG relaxation follows a time-periodic mechanism in which one half-pitch cholesteric layer is formed. The formation of each half-pitch consists of four stages; these stages are discussed in Supplementary Note 5.

The slow–fast relaxation observed for BLG consists of four sigmoid-like steps, and each of which ideally corresponds to the formation of a circular half-pitch, $\frac{p_\infty}{2}$. In particular, the label numbers $B_2$, $B_3$, $B_4$, and $B_5$ in Fig. 1m and Fig. 2 correspond to the circular formation of $\frac{3p_\infty}{2}$, $\frac{4p_\infty}{2}$, $\frac{5p_\infty}{2}$, and $\frac{6p_\infty}{2}$, respectively (see more details in Supplementary Note 4). This behavior is confirmed by simulations and is consistent with the experimental analysis concerning both fibers configuration and trapped defects; however, the simulation prediction of cholesteric layers formation is slightly overestimated. For this reason, the theoretical prediction of relaxation progress, $R$, is slightly higher than the experimental data.

Figure 3 illustrates representative images of the equilibration sequence for CNC fibers, showing that this system relaxes through similar mechanisms, achieving concentric configuration as expected from previous works[17,19] (see Supplementary Note 4). During Phase (I), the order parameter of CNC fibers that are in the para-nematic medium is dramatically dropped to $S_{d,CNC} \approx 0$ at the beginning of relaxation despite their initial order parameter $S_i = 0.6$ (ref. [42]) and as a consequence, isotropic phase emerges from the para-nematic medium, followed by Phase (II), where the chiral front propagates into an isotropic medium (Supplementary Movie 5).

According to the LdG theory, the critical order parameter at the order–disorder phase transition can be considered as $S_c = 0.25$ (ref. [32]), which means that, if $S > S_c$ fibers possess orientational ordering, otherwise fibers lose their correlations and the phase becomes isotropic (see Supplementary Note 7). The simulation results reveal that, during the early relaxation, order parameters of BLG and CNC fibers in the para-nematic mediums have dropped to $S_{d,BLG} \approx 0.33$ and $S_{d,CNC} \approx 0$, respectively. Hence, at the beginning of the relaxation, the fibers in the para-nematic state in the middle of the capillary decrease their order parameter during Phase (I) and the solutions become weakly anisotropic and nearly isotropic for BLG and CNC, respectively. These theoretical predictions are in excellent agreement with POM images because the dark zone in POM images of BLG is slightly blended with a faint reddish color indicating that the fibers therein possess an extremely week orientational ordering and are aligned parallel to the central axis of the capillary, see Figs. 1a–f and Fig. 2c, e, g, i, k. On the other hand, the middle of POM images of CNC is almost dark without any other color denoting that the fibers therein are in the completely isotropic phase, see Fig. 1g–l and Fig. 3c, e, g. Consequently, this supports the assumption we made in the experimental analysis on the dark zones observed in the microscopy images; the dark zones should be considered isotropic areas or para-nematic with extremely low-order parameter, and not nematic nor cholesteric with axes parallel to the light path.

**Understanding the physical origins of relaxation mechanisms**. As above-mentioned, it is found that the relaxation mechanisms can generally be described by two consecutive phases. This section reveals the physics behind the explored mechanisms in terms of the free-energy landscape.

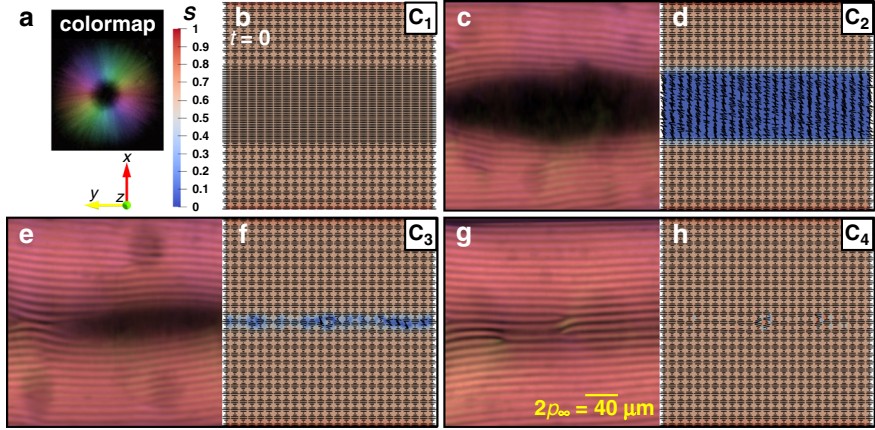

**Fig. 3 Mechanism of smooth dynamic in CNC relaxation. a** The colormap indicating the fiber orientation in the experimental POM panels captured by the LC PolScope device (see Supplementary Note 1 for further information), the blue-to-red spectrum showing the order parameter, $S$, computed by direct numerical simulation (see Supplementary Note 6 for further information), and the coordinate system. **b** Initialization of director field in direct numerical simulation, representing the initial configuration of fibers. **c, e, g** The experimental POM panels corresponding to the $C_2$ to $C_4$ stages shown in Fig. 1n. **d, f, h** The schematic fibers orientation and their order parameter computed by direct numerical simulation in the lateral plane for the $C_2$ to $C_4$ stages shown in Fig. 1n. General Note: the label numbers from $C_1$ through $C_4$ are marked in Fig. 1n.

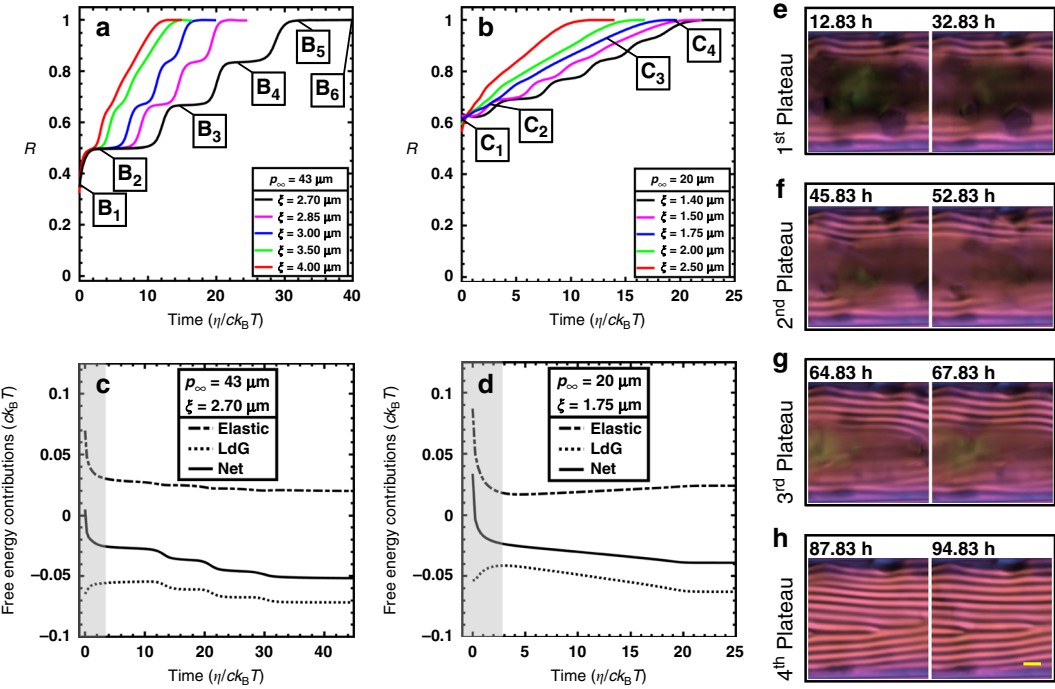

**Fig. 4 The generic mechanism of relaxation dynamics. a, b** Parametric analyses of the normalized relaxation progress, $R$, showing that smooth relaxation dynamics arise, provided coherence length is greater than a threshold. $\eta$ stands for the rotational viscosity coefficient. **c, d** The interplay between homogeneous and elastic free energies, and the resulting net free energy. The gray-hatched zones indicate the period over which the order parameter of the para-nematic phase is dropped, Phase (I), and the remaining white-hatched zones correspond to front propagation, Phase (II). During the former and the latter, as can be appreciated by net free-energy behavior, elastic and homogeneous contributions mainly control net free energy, respectively. Note that all graphs exhibit the spatial-averaged quantities and the panels **c** and **d** correspond to the BLG and CNC relaxations, respectively. **e-h** POM images showing no significant change in the cholesteric zone during each of the four BLG plateaus marked in Fig. 1m. These POM images were experimentally acquired using the LC PolScope device. Scale bar is $p_\infty = 43\,\mu m$.

According to Eqs. (7–9), two factors mainly govern relaxation dynamics: pitch length and coherence length. For a given cylindrical capillary, the impacts of these factors are shown in Fig. 4a, b. The simulations suggest that smooth relaxation can change to slow–fast relaxation and vice versa. For instance, for a given pitch length, smooth relaxation gradually switches to (emerges from) slow–fast relaxation, upon decreasing (increasing) the coherence length.

In liquid-crystalline phases, the coherence length $\xi$ describes the distance over which long-range orientational order varies[49] and is expected to be in the nano-meter range for small-sized rod-shaped mesogens such as 5CB ($\ell_{5CB} = 1.8\,nm$ and $\xi_{5CB} \approx 7\,nm$)[50]. In particular, $\xi$ is directly correlated with length of rod-shaped mesogen[51], and $\xi$ is thus expected to be in the micro-meter range for long fibrous mesogens since their contour lengths are two orders of magnitude longer compared to small-sized mesogens

(e.g. $\ell_{BLG} = 652$ nm and $\ell_{CNC} = 325$ nm). As illustrated in Fig. 4a, b, the parametric analyses shed light on the fact that the observed relaxation dynamics are induced for a micro-meter range of coherence length defined in Eq. (2). The coherence lengths that best fit the experimental data of BLG and CNC shown in Fig. 1m, n are: $\xi_{BL} = 2.7$ μm and $\xi_{CNC} = 1.75$ μm, respectively.

Before proceeding in elucidating the mechanism of cholesteric layers formation, it is important to quantify the impact of order parameter reduction on the free-energy contributions. The homogeneous and gradient elastic free-energy contributions to the net free energy are given by the LdG and FOM theories are[16,25,52–55]:

$$f_{net} = f_h + f_g. \qquad (1)$$

The homogeneous free energy $f_h$ decreases when the order parameter increases from a low to the equilibrium value; therefore, this contribution promotes orientational ordering. The gradient elastic free energy $f_g$ is the energy cost of orientational configuration and is minimum when the ground state is achieved. The decreasing order parameter results in decreasing $f_g$ since Eq. (9) shows that there is a factor of $S$ in all the terms of $f_g$ (see Supplementary Note 8 for more discussion about the impact of order parameter reduction on the free-energy contributions). In partial summary, the order parameter reduction increases the homogeneous contribution and decreases the gradient elastic contribution.

As mentioned earlier, the mechanisms behind the relaxation dynamics are described by two consecutive phases as follows:

*Phase (I)*: early relaxation depicted by the gray-hatched zones in Fig. 4c, d. During this short period, fibers in the para-nematic medium decrease their order parameter. This unexpected behavior can be explained by analyzing the free-energy contributions. The ground state is inherently chiral nematic, thus the para-nematic alignment, which represents unwound chiral helices, gives rise to an excess elastic free energy and, in consequence, rises the net free energy immediately before relaxation starts; see $t = 0$ in Fig. 4c, d. Because the elastic free energy is weighted by the coherence length squared and the para-nematic alignment is unfavorable owing to the concentric ground state, the excess elastic energy is intensified by increasing of coherence length and/or decreasing of pitch length (see Supplementary Notes 8 for the detailed discussion).

To reduce the total excess energy imposed by the initial para-nematic alignment, fibers existing in the para-nematic phase decrease their order parameter to decrease the dominant gradient elastic free energy, although an order parameter reduction is energetically unfavorable for the less costly homogeneous contribution (see Fig. 4c, d). The order parameter of the para-nematic medium thus drops to a low value, $S_d$, shortly after the relaxation starts. Therefore, the more excess elastic energy, the more the order parameter of the para-nematic medium should drop. Taken altogether, the pitch length and coherence length are directly and inversely proportional to the order parameter of the para-nematic medium, $S_d$, respectively (Supplementary Notes 8 and 9).

Since $S_d$ is either close to or lower than the order parameter at the phase transition, $S_c = 0.25$, a dark zone emerges and persists in the POM images while the relaxation continues; for example, Fig. 1a, b, g, h shows that Phase (I) is completed during the first 15–20 min for both BLG and CNC. When the emergence of dark zones in the time-series POM images is completed, Phase (I) terminates. Also, during the early relaxation, i.e., Phase (I), the order parameter reduction largely affects the gradient elastic contributions compared to homogeneous contribution; hence

order parameter reduction results in reducing the net free energy (see Fig. 4c, d).

*Phase (II)*: front propagation illustrated in Fig. 4c, d. As the front propagates inward through the para-nematic medium with the order parameter of $S_d$, the dark zone progressively shrinks, see Figs. 1a–l, 2, 3 and 4e–h. $S_d$, which depends on both the pitch length and the coherence length, is a key factor to determine whether relaxation dynamics become slow–fast or smooth. Numerical simulation supported by experimental observations shows that if $S_d \approx 0$, the para-nematic phase becomes essentially isotropic and the smooth relaxation takes place as discussed for CNC relaxation. As $S_d$ gradually increases, the plateaus are more pronounced. Eventually, relaxation ends up with slow–fast mode in cases that $S_d \approx 0.3$–0.4 like such as BLG relaxation having $S_{d,BLG} = 0.33$ (Supplementary Note 9). As can be seen in Fig. 4e–h, during the plateaus, there is an insignificant change in the formed cholesteric layers, which slows down the relaxation.

Because, as already explained, $S_d$ directly and inversely depends on the pitch length and coherence length, we conclude that a decrease (an increase) in the pitch length and/or increase (decrease) of the coherence length drives the relaxation toward smooth (slow–fast) dynamics (see Supplementary Fig. 12). Finally, regardless of the $S_d$ value, Phase (II) ends by reaching the self-selected ground state of a concentric due to $\frac{D}{p_\infty} > 1$, which is in agreement with previous studies[17,19].

In addition to the coherence length and pitch length, geometric confinement is expected to have a significant impact on relaxation (Supplementary Note 10). This effect is beyond the scope of this work, which is based on a single diameter capillary.

**Properties estimation and relaxation time controllers**. The quantitative agreement between experimental observations and theoretical predictions (Fig. 1m, n) provides a way to obtain quantitative estimates of the rotational viscosity coefficient $\eta$ and coherence length $\xi$. Knowing the coherence length[25,49,53,56–60],

$$\xi = \sqrt{\frac{L_1}{ck_BT}}, \qquad (2)$$

where $k_B$ and $T$ stand for Boltzmann's constant and temperature, it is then possible to calculate the elastic constant $L_1$ (Supplementary Note 11).

Table 1 shows that the equilibrium order parameter computed by numerical simulation is consistent with the results of the analytical equation formulated in previous theoretical studies[25,49]. The elastic constant $L_1$, which is related to Frank's elastic constants[52], is in the range used in other studies[24,61–63]. To the best of authors' knowledge, there has not been any report in the literature concerning rotational viscosity coefficients of BCLLCs. There, however, exists a few studies where rotational viscosity coefficients have experimentally been measured for large molecular-weight LCs, reporting $2.8 \times 10^5$P and $9.9 \times 10^5$P for PBLG and mesogenic polyesters, respectively[35,36]. As justified in ref. [36], in comparison to thermotropic LCs, BCLLCs possess a large rotational viscosity coefficient because the relaxation time of thermotropic LCs is in range of milliseconds while BCLLCs ranges from hours to days.

Having rotational viscosity coefficients of BLG and CNC estimates, we can answer whether lower rotational viscosity of CNC is the reason behind its fast relaxation dynamic. The rotational viscosity $\gamma$ is[25,32]

$$\gamma = \eta \left(1 - S^2\right)^2 \qquad (3)$$

and increases with decreasing $S$. The results shown in Table 2 confirm that the rotational viscosity of BLG is larger than that of

CNC throughout the relaxation in the order of $\frac{\gamma_{CNC}}{\gamma_{BLG}} \approx 0.57 - 0.64$. Therefore, lower rotational viscosity of CNC speeds the relaxation.

The free-energy driving force also contributes to the relaxation kinetics as the phase ordering rate is (see section "Direct numerical simulation" and Eq. (5))

$$\text{ordering rate} = \frac{\text{free energy driving force}}{\text{rotational viscosity}}. \qquad (4)$$

The difference between initial and equilibrium net free energy is representative of the free-energy driving force, $\Delta F$. Based on Fig. 4c, d, we conclude that the free-energy driving force for relaxation of CNC is also larger than that of BLG, $\frac{\Delta F_{CNC}}{\Delta F_{BLG}} = 1.28$. Consequently, CNC relaxation is faster because of two synergetic effects: (1) lower rotational viscosity and (2) higher free-energy driving force.

## Discussion

We integrated theoretical and experimental approaches to elucidate the relaxation dynamics (i.e. the spontaneous PN–N* transition) of cylindrically confined BCLLCs with planar anchoring. The analyses of the relaxation dynamics of BLG and CNC reveal that, despite the noticeable similarities between these two biological mesogens, their relaxation behaviors differ considerably. Specifically, BLG slowly relaxes through a slow–fast dynamic while CNC relaxation is fast with a smooth dynamic (Fig. 1). Given the success of liquid-crystal continuum theory in the prediction of relaxation dynamics, we employed standard models to rationalize the mechanisms of relaxation in cylindrical confinement governed by a spontaneous interplay of chirality and viscoelasticity. The following two points should be taken into account regarding the modeling and simulation implemented in this study. First, the results presented here are based on the strong planar anchoring supported by experimental observations. Previous studies[64–66] have shown that the effect of anchoring can be significant. Regarding our study, we also anticipate that different anchoring may affect relaxation dynamics; however, it requires future studies. Second, the hydrodynamics is reasonably neglected from the modeling (see section "Direct numerical simulation"). This reasonable simplifying assumption leads two deviations in the simulation results from experimental observations: (1) simulation predicts that the chiral fronts almost evenly propagate inward while the POM images show that the front propagation is slightly uneven (see Figs. 2 and 3). For this reason, the simulations can overestimate or underestimate $R$ over the time evolution (see Fig. 1m, n). (2) The defect motion cannot be fully tracked since the weak orientation-induced backflow is not considered. However, this insignificantly affects $R$.

The relaxation generally takes place through two phases (see Figs. 2 and 3). Phase (I): this phase describes the early relaxation. The para-nematic alignment existing at the beginning of relaxation leads to increase of the elastic free energy $f_g$ as the system is away from the equilibrium (see Fig. 4c, d). The excess energy of $f_g$ is then relieved (decreased) to some extent as the order parameter of the unwound medium decreases to a low value, $S_d$ (Supplementary Note 8). Even though the order parameter reduction also increases the homogeneous contribution, $f_h$, the net free energy, $f_{net}$, decreases because $f_{net}$ is significantly affected by $f_g$ compared to $f_h$ during Phase (I) (see Fig. 4c, d). The dark zones emerging in the middle of the POM images are thus attributed to the low value of $S_d$. $S_d$ is directly and inversely affected by the pitch length and the coherence length, respectively (Supplementary Notes 8 and 9). Phase (II): during this phase, the chiral front propagates through the low-order-parameter medium. As the front propagates, the cholesteric layers are formed and their order parameter evolves

from a low value, $S_d$, into the equilibrium value $S_e$ (Supplementary Movies 4 and 5). As can be seen in Fig. 4c, d, during Phase (II), $f_h$ dominates the net free energy; for this reason and the fact that $f_h$ decreases upon ordering, cholesteric layers formation leads to further decrease of net free energy. Besides, we found that the type of relaxation dynamics can be determined by the value of $S_d$; the propagation through the significantly low-order-parameter para-nematic medium (i.e. the isotropic medium), $S_d \approx 0$, results in a smooth relaxation dynamic, and higher values of $S_d$ lead to slow–fast dynamics (Supplementary Notes 8 and 9).

Lastly, we proposed a systematic framework to estimate elastic constants and rotational viscosity. Besides the reasonable estimations shown in Tables 1 and 2, we showed that two factors control the relaxation time: rotational viscosity and free-energy driving force. Therefore, our study introduces a new general methodology where the experimental mapping of cholesteric pitch evolution over time, benchmarked by numerical simulations, allows recovering fundamental properties of the filamentous chiral bio-colloid under investigation, some of which are notably difficult to obtain via alternative methods.

## Methods

**Preparation of BLG cholesteric bulk.** Amyloid fibrils were prepared by heating a pH 2, 2 wt% β-lactoglobulin solution at 90° for 5 h. Mature amyloids were shortened and homogenized using an immersion mixer for 1 min. The amyloid solution was purified with dialysis against pH 2 milliQ for 5 days using semipermeable membranes (MWCO 100 kDa) and then up-concentrated to reach the isotropic nematic coexistence regime with reverse osmosis against 5 wt% 35 kDa PEG solution using semipermeable membranes (MWCO 6–8 kDa). The solution was equilibrated for several weeks until complete phase separation was reached (see Supplementary Note 12).

**Preparation of CNC cholesteric bulk.** Cellulose nanocrystal solution was prepared by mixing 2.5 wt% freeze-dried CNC (FPInnovations) in milliQ water. To disperse CNC, the solution was ultra-sonicated for 2 min and centrifuged for 20 min at 12,000 r.c.f. The supernatant was collected, and the solution equilibrated for several days until a macroscopic phase separation was achieved (see Supplementary Note 12).

**Preparation of BLG and CNC samples for optical microscopy.** The cylindrical capillaries (VitroTubes, Vitrocom) of inner diameter 260 μm were filled with bulk cholesteric phase (BLG: 2.8 wt% and CNC: 2.7 wt%) through capillary suction, by immersion of the capillaries into the anisotropic solutions (see Supplementary Note 12). The capillary tubes were immediately sealed with UV glues to avoid evaporation.

**PolScope.** The relaxation dynamic was recorded using an optical microscope combined with an LC PolScope universal compensator, allowing the quantitative analysis of birefringence[44]. In particular, after focusing on a region of interest in the middle of the cuvettes, time-series images with a specific time step were collected and analyzed with PolScope. The device is used for birefringence imaging to analyze with high sensitivity and high resolution the spatial and temporal variations of the phase delay in anisotropic materials and to provide pixel-by-pixel information of local optical anisotropy together with a mapping of the slow axis orientation of birefringence regions.

**Direct numerical simulation.** The capillaries are exclusively filled with the fully cholesteric phase existing at the constant equilibrium concentration and then immediately sealed (see the section "Preparation of BLG and CNC samples for optical microscopy"); hence, the concentration of the system remains a constant equal to the upper binodal concentration throughout the relaxation time. Moreover, during the relaxation, it is impossible that the system spontaneously splits into two distinct concentrations, meaning that the low-order-parameter phase (dark zone) and high-order-parameter phase (fingerprint zones) are depleted and concentrated, respectively. This can be proven by *proof by contradiction*; based on the mass conservation law, if the dark zone becomes fiber-lean then the fingerprint zones become more fiber-rich. Under this condition, the fingerprint zones sharply approach the gel phase. However, time-series POM images do not point to such condition (see Supplementary Movies 1 and 2). Altogether, the relaxation dynamics presented in this study takes place in a thermodynamically closed system at constant concentration field without appreciable variation. Owing to the insignificant mass exchange, the mass continuity equation is reasonably neglected.

The essential physics which is necessary to be taken to account is the spatio-temporal dynamics of fibers orientation. The liquid-crystalline ordering is described by the second-order symmetric traceless tensor, $\mathbf{Q}(\mathbf{x}, \mathbf{t})$, whose eigenvalues and eigenvectors correspond to the order parameters and the macroscopic orientation of mesogens, respectively. The spatio-temporal orientational relaxation follows the time-dependent Ginzburg–Landau model whereby $\mathbf{Q}(\mathbf{x}, t)$ is thermodynamically allowed to evolve toward equilibrium[25].

$$\underbrace{\frac{\partial \mathbf{Q}}{\partial t}}_{\text{ordering rate}} = -\frac{1}{\gamma} \underbrace{\left(\frac{\delta F_{\text{net}}}{\delta \mathbf{Q}}\right)^{[s]}}_{\text{driving force}}, \tag{5}$$

$$\frac{\partial \mathbf{Q}}{\partial \tilde{t}} = -\frac{1}{\left(1 - 3\text{Tr}(\mathbf{Q}^2)/2\right)^2} \left(\frac{\delta \tilde{F}_{\text{net}}}{\delta \mathbf{Q}}\right)^{[s]}. \tag{6}$$

Equation (6) is the dimensionless form of Eq. (5). The actual time, $t$, and the dimensionless time, $\tilde{t}$, are related by $t = \frac{\tilde{t}\eta\xi^2}{L_1}$. [s] and Tr indicate symmetric traceless and trace tensorial operations, respectively. $\delta\tilde{F}_{\text{net}}/\delta\mathbf{Q}$ shows the functional derivative of total dimensionless free energy, $\tilde{F}_{\text{net}}$, with respect to $\mathbf{Q}$-tensor. $\tilde{F}_{\text{net}}$ is comprised of two contributions; homogeneous effect of phase ordering, $\tilde{f}_{\text{h}}$, described by the LdG theory and the gradient effect of elasticity, $\tilde{f}_{\text{g}}$, given by the FOM theory[25,32,49]:

$$\tilde{F}_{\text{net}} = \int_{\tilde{V}} \left(\tilde{f}_{\text{h}} + \tilde{f}_{\text{g}}\right) d\tilde{V}, \tag{7}$$

$$\tilde{f}_{\text{h}} = \frac{1}{2}\left(1 - \frac{U}{3}\right)\text{Tr}(\mathbf{Q}^2) - \frac{1}{3}U\text{Tr}(\mathbf{Q}^3) + \frac{1}{4}U\left(\text{Tr}(\mathbf{Q}^2)\right)^2, \tag{8}$$

$$\tilde{f}_{\text{g}} = \frac{1}{2}\left(\frac{\xi}{h_0}\right)^2 \left[\left[\tilde{\nabla} \times \mathbf{Q} + 4\pi\left(\frac{h_0}{p_\infty}\right)\mathbf{Q}\right]^2 + \alpha\left[\tilde{\nabla} \cdot \mathbf{Q}\right]^2\right] \tag{9}$$

$U$ denotes the nematic potential parameterized as $U = \frac{3c}{c^*}$ in which the asterisk shows the order–disorder phase transition. $h_0$ is an arbitrary macroscopic length scale which is representative of sample size. $\alpha$ is the anisotropic ratio defined as $\alpha = \frac{L_2}{L_1}$ in which $L_i$ are elastic constants[52]. In our present simulations, we found that the contribution of $\left[\tilde{\nabla} \cdot \mathbf{Q}\right]^2$ was not significant.

As can readily be appreciated through the POM images, both BLG and CNC have strong planar anchoring with the inner capillary surface. To capture this state, the governing equations are subjected to

$$\mathbf{Q}(\mathbf{x},t)|_{\text{surface}} = \begin{bmatrix} -1/3 & 0 & 0 \\ 0 & 2/3 & 0 \\ 0 & 0 & -1/3 \end{bmatrix}. \tag{10}$$

Equation (10) indicates that fibers located on the bounding surface are aligned parallel to the central axis of the capillary with strong anchoring, $S = 1$. Furthermore, as shown in Figs. 2b and 3b, fibers existing in the bulk are initially aligned along the central axis of the capillary; thus, the initial condition is set as

$$\mathbf{Q}(\mathbf{x},t=0)|_{\text{bulk}} = S_i \begin{bmatrix} -1/3 & 0 & 0 \\ 0 & 2/3 & 0 \\ 0 & 0 & -1/3 \end{bmatrix}, \tag{11}$$

where $S_i$, as mentioned earlier, stands for initial order parameter and equals to 0.6 (ref. [42]). Readers are referred to Supplementary Note 13 for a detailed discussion on the numerical simulation methods.

Lastly, the relaxation progress, $R(t)$, can also be parameterized in terms of $\mathbf{Q}$-tensor.

$$R(t) = \frac{\int_A \frac{\mathbf{Q} : \nabla \times \mathbf{Q}}{\mathbf{Q} : \mathbf{Q}} dA}{\left(\int_A \frac{\mathbf{Q} : \nabla \times \mathbf{Q}}{\mathbf{Q} : \mathbf{Q}} dA\right)_e}, \tag{12}$$

where subscript e indicates equilibrium, and integration is over the lateral plane in which the striped pattern (twisting monodomain) is formed, see Fig. 1o and Supplementary Note 14.

In the modeling used in this study based on Eqs. (5–11), there is no external stress (see Supplementary Note 15). Furthermore, hydrodynamics is reasonably negligible because of three reasons. First, as already explained, the orientational relaxation takes place in the closed system in which there does not exist any external velocity driving force such as pressure difference, gravity (the filled capillaries were kept horizontally throughout the experiments), moving surface, and so on. Second, as above-mentioned, there is no appreciable concentration gradient in the system; hence, the hydrodynamics induced by mass transport becomes negligible. Third, the self-generated transient bulk convection is insignificant owing to following reasons: the viscosity of cholesteric permeation flow is extremely large[31], the rotational viscosity of BLG and CNC is also considerably large (see Table 2), the essentially vanishing Ericksen number (flow-to-elasticity ratio)[67], the solid-like behavior along the cholesteric helix axis[68], and

the a posteriori validation and high fidelity of the predictions with experimental data.

**Reporting summary**. Further information on research design is available in the Nature Research Reporting Summary linked to this article.

## Data availability
The authors declare that the main data supporting the findings of this study are available within the article and its Supplementary Information files. Extra data are available from the corresponding author upon request.

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

## Acknowledgements
S.A.K. acknowledges financial support from the McGill Engineering Doctoral Awards (MEDA) program. A.D.R. is thankful to McGill University for financial support through the James McGill Professorship appointment. This research is supported by a Sinergia grant from the Swiss National Science foundation (CRSII5_189917/1).

## Author contributions
S.A.K. and M.B. contributed to the theoretical and experimental parts, respectively. R.M. and A.D.R. designed and supervised the research. All authors discussed the results, contributed to writing, and agreed with the contents.

## Competing interests
The authors declare no competing interests.
