## [Peer Review File · Nature Communications]

This is an important contribution that provides insights into some of the poorly understood aspects of physical behavior of biologically derived cholesteric liquid crystals. The combination of experiments and modeling is a powerful tool to address this behavior. Considering that the results have important insights for many branches of physics, technology, biology and materials science, I believe the manuscript meets the high standards and requirements for NatComm publications in terms of broad interest and novelty. While I think the manuscript should be eventually published, I have some suggestions that I believe are important for authors to consider and account for.

The title of article might be more general than the scope of work would suggest. For example, cholesteric liquid crystals formed by persistence-length 50nm DNA fragments likely would exhibit different behavior and I see no reason to suggest that the findings described here would be applicable to all biologically derived cholesterics - authors should modify the title to address this.

Authors use Frank-Oseen-Mermin in the main text and just Frank-see in SI - the reader may be confused and, as a minimum, authors should carefully describe the differences and what they actually. The literature review and discussion seem to be missing the discussion of potential technological implications of this work. For example, control of equilibrium and metastable states in the forms of cholesterol and paranematic structures is important for forming retarder films and cholesterol color filters [ACS PHOTONICS 5, 2468-2477 (2018)] as well as for plasmonic mesostructured materials [ADVANCED MATERIALS 26, 7178-7184 (2014)].

I found some of the SI movies and figures very helpful in following this work - could authors consider moving SI figures S1 and S2 to the main text? I believe the format of NatComm articles allows for this.

$$-\nabla f + \nabla \cdot \left\{ \frac{\partial f}{\partial \nabla \mathbf{Q}} : (\nabla \mathbf{Q})^T \right\} = -\frac{\partial f}{\partial \mathbf{Q}} : (\nabla \mathbf{Q})^T + \left(\nabla \cdot \left(\frac{\partial f}{\partial \nabla \mathbf{Q}} \right) \right) : (\nabla \mathbf{Q})^T \quad (\text{S30})$$

Comparing eqs.(S25, S30) results in

$$\gamma \left(\frac{\partial \mathbf{Q}}{\partial t} \right) : (\nabla \mathbf{Q})^T = -\nabla f + \nabla \cdot \left\{ \frac{\partial f}{\partial \nabla \mathbf{Q}} : (\nabla \mathbf{Q})^T \right\} = \mathbf{H} : (\nabla \mathbf{Q})^T \quad (\text{S31})$$

Additionally, we know that the mechanical bulk total elastic stress tensor \mathbf{T} is

$$\mathbf{T} = f \mathbf{I} + \mathbf{T}^E \quad (\text{S32})$$

$$\mathbf{T}^E = -\frac{\partial f_g}{\partial \nabla \mathbf{Q}} : (\nabla \mathbf{Q})^T \quad (\text{S33})$$

where \mathbf{T}^E is the Ericksen stress tensor, and the pressure $p = -f$ is minus the free energy density. Then, the bulk vector force balance eq.(S24) reads:

$$-\gamma (\nabla \mathbf{Q}) : \left(\frac{\partial \mathbf{Q}}{\partial t} \right) = \nabla \cdot \mathbf{T} \quad (\text{S34})$$

The total stress tensor, eqs.(S32, S33), carries information on f and $-\frac{\partial f_g}{\partial \nabla \mathbf{Q}} : (\nabla \mathbf{Q})^T$. The vector mechanical force balance, eq.(S34), carries less information than the original tensorial \mathbf{Q} -tensor equation, eq.(S23) since the former is a vector equation and the latter a tensor equation. Consequentially, when we simulate \mathbf{Q} -tensor process in the absence of mass velocity ($\mathbf{v}=0$), the \mathbf{Q} -tensor model carries more information by just focusing on the tensorial variational derivative of the free energy instead of the vectorial divergence of the stress.

The total elastic stress \mathbf{T} is always present $\mathbf{T} \neq \mathbf{0}$ and, in consequence, the elastic force \mathbf{F} which is the divergence of the total bulk elastic stress is also nonzero, $\mathbf{F} = \nabla \cdot \mathbf{T} \neq \mathbf{0}$ as \mathbf{F} is balanced by $\gamma \left(\frac{\partial \mathbf{Q}}{\partial t} \right) : (\nabla \mathbf{Q})^T + \mathbf{F} = \mathbf{0}$. To sum up, there is no applied external stress in this experiment or in this model, and there is only internal stress (\mathbf{T}) properly embedded in the modeling used in this study."

Response to Comment (2): In addition to the significant findings described in the previous comment, we devised an experimental-theoretical protocol whereby the orientational relaxation dynamics is harnessed to estimate the hard-to-measure quantities including viscoelastic properties; namely, coherence length, order parameter, elastic constant, rotational viscosity coefficient, and rotational viscosity. We believe that the proposed methodology will become a new widely-used standard and attract a broad audience because of the following reasons:

1. Estimation of viscoelastic properties contributes to relaxation dynamics and the nematodynamics field which itself encompasses numerous practical and theoretical applications, see Lines 49-52 in manuscript.

Authors' Response to the Referees' Reports

2. Estimation of viscoelastic properties have been a long-lasting challenge in the biological, low-molar mass and polymeric liquid crystal communities, see Lines 53-58 in manuscript.
3. The proposed approach is both straightforward and systematic (see Figure S10).
4. The salient feature of the proposed method is that it does not require applying magnetic or electric field, allowing measurement of viscoelastic properties of low diamagnetic or dielectric LCs whose viscoelastic properties (e.g. rotational viscosity) are difficultly measurable by conventional method based on applying magnetic or electric fields, see Lines 58-63 in manuscript.
5. The proposed method has been validated for two important biological cholesteric LCs; BLG and CNC, see Lines 398-417 in manuscript.

Action for Comment (2):

In Manuscript, Lines 50-52: “Another importance of viscoelastic properties lies in their usages in LC rheology nematodynamics, and flow-processing of fibers and films^{1,30-36} such as the biomimetic material design through coating¹¹.”

In Manuscript, Lines 402-403: “see the Supplementary Note S9 for the flow chart indicating the systematic framework to estimate the properties.”

In SI, Line 230:

“Note S9. Viscoelastic Properties Estimation Flow Chart

is of particular interest for the BCLLCs. In addition to validating the estimated properties acquired by the proposed framework, the physical factors controlling the relaxation time are then discussed.”

In Manuscript, Line 104: “Polarized optical microscopy (POM).” has changed to “**Distinct Relaxation Dynamics**”

In Manuscript, Line 304: “Relaxation dynamics mechanism” has changed to “**Understanding the physical origins of relaxation mechanisms**”.

In Manuscript, Line 397: “Material properties estimation and identifying factors controlling the relaxation dynamics” has changed to “**Material properties estimation and factors controlling the relaxation time**”.

Lines 239-242 of the first submitted manuscript were deleted: “The simulation results show that fibers in the para-nematic state decrease their order parameter during Phase (I), suggesting that the dark areas appearing in the POM images refer to the isotropic state for CNC and the para-nematic state with extremely low order parameter for BLG”

Lines 251-258 of the first submitted SI were deleted: “The mechanism of slow-fast relaxation can be divided into two phases explained below.

Phase (I): Fibers immediately lose their order parameter on account of relieving excess elastic energy induced by initial unwound helices. Approximately, the order parameter of the para-nematic medium, which is initially set as $S_i=0.6$, drops to $S_d=0.33$. Note that the loss of order parameter takes place in a few first frames.

Phase (II): Front propagation through a para-nematic medium with low order parameter, $S_d=0.33$. This phase itself comprises of four key stages elaborated in Supplementary Note S2 and Movie S4.”

Comment (4): The details of the simulation results are put in the supplementary information, which is fine. However, I think that the description of the experimental samples and of the numerical model are important and missing.

Response to Comment (4): This comment has been fully addressed through the Comments (6), (7), (12)

Action for Comment (4): Please, see Action for Comments (6), (7), (12)

Comment (5): The authors claim that both experimental systems are similar. They should provide from the beginning a table supporting this claim. What are the sizes of the rods, their critical concentration, their natural twist, in short their known liquid crystalline properties (having a factor 2 in length and pitch seems to make them quite different in fact).

Response to Comment (5): We have included the length distribution extracted from AFM measurements and elaborated the similarity of BLG and CNC in detail in the revised version.

Action for Comment (5):

In Manuscript, Lines 69-76: “In particular, we characterize the relaxation behavior of BLG and CNC; these BCLLCs are aqueous solutions of semi-rigid rod-like bio-colloidal LCs characterized by a long pitch length ($p_{\infty, \text{BLG}}=43\mu\text{m}$ and $p_{\infty, \text{CNC}}=20\mu\text{m}$), high aspect ratio ($\ell_{\text{BLG}}=652\text{nm}$ and $\ell_{\text{CNC}}=325\text{nm}$, $D_{\text{BLG}}=4\text{nm}$, $D_{\text{CNC}}=4.6\text{nm}$, $(\ell/D)_{\text{BLG}}=163$, and $(\ell/D)_{\text{CNC}}=71$),

solution equilibrated for several days until a macroscopic phase separation was achieved (see Supplementary Note S10).

Preparation of BLG and CNC samples for optical microscopy

The cylindrical capillaries (VitroTubes, Vitrocom) of inner diameter $260\ \mu\text{m}$ were filled with bulk cholesteric phase (BLG:2.8 wt% and CNC:2.7 wt%) through capillary suction, by immersion of the capillaries into the anisotropic solutions (see Supplementary Note S10). The capillary tubes were immediately sealed with UV glues to avoid evaporation.”

In SI, Lines 232-239:“

Note S10. Sample Preparation

Figure S11. Photograph of phase-separated BLG or CNC dispersion sandwiched between crossed polarizers. From the top to the bottom, phases are Air, Isotropic and Cholesteric, respectively. Note BLG and CNC exactly look like each other in the shown container.

The capillaries were filled with the birefringent solution which is the bulk cholesteric. According to the thermodynamics of phase equilibria, the bulk cholesteric is at a constant concentration equal to the upper binodal curve.”

Comment (7): The microscope images are treated using a device called PolScope, with which not everyone in the field is familiar. It should be specified, that only the component of the director in the plane of the image is detected. In addition, it should be specified whether the image is an average over the thickness of the sample or if it shows a slice (of which thickness) in the middle of the sample.

Response to Comment (7): We thank the referee for raising this important comment. We have added a dedicated section elaborating the LC PoleScope device in the “**Methods**” section. With regard to the two specific questions, we have provided elucidation in the “**Distinct Relaxation Dynamics**” section in the revised manuscript.

Action for Comment (7):

In Manuscript, Lines 105-109: “... by PolScope⁵². The analysis allows extrapolating the average of fibers' orientation over the thickness of capillary tube and the 2D fibers' orientation is represented according to the colormap shown in Figure 1(a); for example, the pink color which is ubiquitous in the POM images indicates that fibers are aligned parallel to the central axis (y-

The experimental time-series POM images showing relaxation of BLG for one week.

Movie S2. CNC Relaxation

The experimental time-series POM images showing relaxation of CNC for one day."

Comment (10): I am surprised, that the authors seem to oppose a fast and a smooth relaxation (line 19 for instance). The steps in the relaxation arise because the process is blocked, so it is natural that, when the process has steps (is not smooth) it also takes longer (relaxation is blocked).

Response to Comment (10): We agree with referee's comment. We have considered this point in the revised manuscript.

Action for Comment (10):

In Manuscript, Lines 20-21: ", and of CNC, characterized by fast equilibration obtained through a smooth relaxation dynamic."

Comment (11): The choice of this definition of R deserves more justification, because it looks like in the images (and in the movies) that the pitch is actually constant in time where the cholesteric phase is formed and not defined in the isotropic zone.

Response to Comment (11): Thank you for making this valuable comment. We have provided a thorough interpretation on the concept of R in the revised SI, and more explanation in the revised manuscript.

Action for Comment (11):

In Manuscript, Lines 129-131: "This quantity, R, provides insight into the amount of space occupied by the cholesteric phase at any given time in the system. The relaxation progress, R, ranges from 0 to 1; for R=0, there is no chiral nematic phase in the system, and when R=1,"

In SI, Lines 286-321: "Now that the normalized relaxation progress, eq.(S22) or eq.(9), is formulated and our experimental-theoretical approach reveals that the dark zone in the POM images is a para-nematic phase with the order parameter of nearly 0 to 0.4, the conceptual understanding of this quantity (R) deserves more discussion. As explained in the paper, when R is computed via the discretization of the time-series POM images, R is taken to be 0 and 1 in dark and fingerprint partitions, respectively. R=0 is equivalent to q=0 signifying that the phase is para-nematic and R=1 indicates $q=q_{\infty}$ representing that the phase is cholesteric. In the case of BLG relaxation (i.e. Figure 2), the R distribution theoretically becomes as what is shown in Figure S13.

Figure S13. A representative snapshot depicting the BLG relaxation and the corresponding R , q , S , and director field (\mathbf{n}).

In the case of CNC, the para-nematic phase loses its order parameter to an extremely low value, $S \sim 10^{-2}$, see Figure S14.

Figure S14. A representative snapshot computed for the CNC relaxation and the corresponding R , q , S , and director field (\mathbf{n}).

The actual fibers' orientation can be understood in light of two factors. First, the uniaxial director field, \mathbf{n} , representing the average fibers' orientation. Second, the uniaxial order parameter, S , describing the strength of fibers alignment around \mathbf{n} . Fibers lie perfectly parallel to \mathbf{n} if $S=1$ and the resulting phase becomes more crystal-like. In the case of $S \approx S_c \sim 0.7$, fibers retain both fluidity and crystallinity (orientational order), corresponding to the liquid-crystalline phase. Finally, $S < S_c$

indicates that the actual fibers’ orientation can be less aligned around \mathbf{n} ; hence, the phase possesses more fluid-like characteristics rather than crystalline ones^{3,6,8,12-14}. As explained in the Supplementary Note 5, there is no unanimous agreement on the S_c value; however, $S_c=0.25$ suits for the theory used in our study. Accordingly, wherever $S < S_c$, the fibers’ orientation is randomly visualized in order to emphasize the concept of critical order parameter and the fact that orientational ordering is weak, see Figure S15.

Figure S15. The snapshot visualized for the Figure S14.

Wherever $S < S_c$, the phase can also be called isotropic due to the fact that the correlation existing among fibers is insignificant. However, the distinguishing point that should be taken into account is that the concentration of the isotropic phase is still at the upper binodal curve, which is unequivocally greater than the critical order-disorder transition, see “Direct Numerical Simulation” section for discussion on the concentration field in the present study.”

Comment (12): The numerical model seems to work well. The reader would like to know a little more about the details of the model (in the main text or in supplement). Is it fully 3D? Is it a finite element model? Is it solved in tensorial or director+order parameter formulation? Is the concentration field taken into account? I would assume that the "isotropic" region is depleted while the cholesteric region is more concentrated.

Response to Comment (12): We greatly appreciate the referee’s constructive comment. We have carefully all points mentioned.

Action for Comment (12):

In Manuscript, Lines 522-537: “The capillaries are exclusively filled with the fully cholesteric phase existing at the constant equilibrium concentration and then immediately sealed (see the section “Preparation of BLG and CNC samples for optical microscopy”); hence, the concentration of the system remains a constant equal to the upper binodal concentration throughout the relaxation time. Moreover, during the relaxation, it is impossible that the system

simulations. In this regard, the governing equations, eqs.(5,6), along with axillary conditions, eqs.(7,8), and parameters tabulated in Table S2 were implemented in the General PDE solver of COMSOL Multiphysics 5.3a on our in-house supercomputer. Note that the model used in this study is a tensorial equation, leading to five second-order time-dependent nonlinear coupled PDEs. Time stepping was executed using Backward Differentiation Formula (BDF) with varying orders from one (known as the backward Euler method) to five.

Table S2. Parameters used in the present study to capture the dynamics of orientational relaxation.

Parameter	Value
U	4.5
α	19

All simulations are performed in a 3D computational domain except those which are intended for comparison with the POM images. Figure S12 shows the geometries and the simulation type used in this study.

Simulation Results Geometry used

Figure 1(c,d)	2D
Figure 2*	2D
Figure 3*	2D
Figure 4(a-d)	2D
Figure S3	3D
Figure S4	3D
Movie S3	3D
Movie S4	3D
Movie S5	3D
Movie S6	3D

**In each panel, the right side is simulation*

Figure S12. Summary of the geometries used in the present study. ”

In SI, Lines 323-352: please see Action for Comment (1)

Comment (13): The manuscript could benefit from a topological consideration: starting from a homogeneous state, is it possible to change continuously to the final structure without generating or propagating defects (disclinations). Can the center of the capillary tube be considered as a

giant disclination?

Response to Comment (13): This paper focuses on the kinetics and dynamics of the paranematic-to-cholesteric transition (PN-N*), including the time evolution of chirality and time-dependent mechanisms involved in the interfacial motion. Since the focus of the experiments is time evolution of a phase transformation, we use energy-based tools to accurately describe the phenomena.

Regarding “The manuscript could benefit from a topological consideration: starting from a homogeneous state, is it possible to change continuously to the final structure without generating or propagating defects (disclinations).”, for the materials and conditions under consideration we clearly observe and predict defects. This study focuses on lyotropic bio-colloidal liquid crystals (LBCLCs), e.g. BLG, CNC, collagen, spider silk, which are similar to lyotropic polymeric LCs (LPLCs). The close relations in the material properties and defect physics (defect types, nucleation, annihilation, interactions, frustration, confinement) between LBCLCs and LPLCs were discussed in:

- Rey, Alejandro D. "Liquid crystal models of biological materials and processes." *Soft Matter* 6.15 (2010): 3402-3429.

Defects in these materials are invariably observed and due to high viscosities and little mobility, they remain in the system even if they can annihilate with others or escape through the boundaries or in the case of chargeless disclination loops shrink and disappear. These expected issues are discussed in the literature and are not the focus of this paper. The presence and role of defects through the PN-N* transition was discussed in the paper and carefully characterized in Figures 1(a,b), 2, 3, movies S1-S2, Supplementary Note S2, and Lines 182-210 of the manuscript.

For which other material and geometric conditions (different from ours) avoid defect nucleation and propagation starting from an initial paranematic phase with axial director orientation and azimuthal anchoring one could think of changing the internal and external length scale ratios that control energy budgets and the time scale ratios. Identifying these material and geometric conditions that affect energy and time scales and could result in total absence of

Comment (14): I also draw the attention of the authors, that this growth of the cholesteric phase from the wall could perhaps be modeled in a renormalized/self-consistent way: the growth the next layer is similar to the growth of the first layer inside a tube of smaller diameter.

Response to Comment (14): In the study presented here, we have already revealed the generic mechanisms and the physical origin of relaxation dynamics by utilizing the standard well-established continuum theory (i.e. LdG and Frank-Osenn-Mermin). The idea proposed by the referee is very interesting for us. We have been working to theoretically prove a recursive mathematical model describing the relaxation dynamics. However, we strongly believe that development, implementation, and validation of such mathematical model deserves to be dedicatedly considered in a separate study. Hence, this subject is out of scope of our current study and might not also be attractive for the broad Nature Communications audiences.

Reviewer #2 (Remarks to the Author):

Referee: The main aim of this paper is to study the relaxation dynamics of two chiral lyotropic liquid crystals (BNG and CNC) confined into a cylindrical capillary upon a quenched into a para-nematic state. This is carried out through a nice combination of experiments and simulations that allow both to understand the underlying physical mechanisms of the relaxation process and to design a non-standard (but promising) procedure to measure viscoelastic properties of chiral liquid crystals (CLC) that have very low diamagnetic and dielectric response. For these reason the unwinding procedure of the ground state helices (i.e. the quench to the non equilibrium para-nematic state) is not performed via the application of external magnetic or electrical fields but relies on the onset of flow aligning mechanism during the filling of the capillary.

By taking microscopic images of the LC samples during their relaxation process, the authors have been able to show that the two above cited systems relax over different time scales and following very diverse dynamical pathways: for the BNG relaxation develops rather slowly (it takes days to reach equilibrium again) and follows a sequence of fast-slow step-process characterised by stalling periods; the CNC system relaxes instead more rapidly following a smoother evolution. This intriguing difference is then rationalised by simulating a well known mean field model of CLC based on the minimisation of a free energy that depends on physical parameters of the CLC such as the elastic constant, the coherence and pitch lengths and the rotational viscosity. After having verified that simulations faithfully describe the time evolution of the experimental observations the authors performed an extensive numerical analysis of the relaxation process as a function of the coherence length. This study allowed to pin point that relaxation is mainly governed by the interplay of chirality and viscoelasticity.

An interesting finding is that the difference in the dynamical pathway is mainly due to the competition between the para-nematic state that increases the elastic free energy and the isotropic state that increases the bulk free energy of the CLC. This competition results into the formation of a either weakly order paramagnetic phase (CNC) or to a fully isotropic phase (BNG). This strongly suggests that the dynamical pathway is mostly determined by the dominance of either of these phase in the early stage of relaxation. s

Finally it is of great interest the suggested systematic framework to estimate elastic constants and rotational viscosity based on a virtuous combination of numerical simulations of relatively simple theoretical models of CLCs and experimental observations of real systems that do not imply the use of external magnetic or electric fields.

There remain many fundamental issues about how confinement can affect the relaxation pathways of CLCs and how to exploit the richness and complexity of these viscoelastic responses for a design of new meso-materials with non trivial dynamical properties.

This paper should therefore be of wide interest within the wide community of soft matter and biological physics and could attract the Nature Communications audience.

The manuscript has however some points that needs to be clarified/strengthen before being considered publishable in Nat Com.

Reply: The authors would like to thank the referee very much for the positive assessment and the constructive comments for improving the manuscript. We have carefully revised our manuscript and addressed all points one-by-one as follows. Implementation of the given comments enhances the quality of the revised manuscript.

Comment (1): The simulations are based on a de Gennes-Landau (mean field) model of the CLCs where the relaxation dynamics is essentially a minimisation procedure of a given free energy. In this description hydrodynamic modes are fully neglected. However this approximation does not seem to affect the shown numerical results that are in good agreement with the experimental data. How the authors can explain this finding ? Does it really mean that in this relaxation process (that sometimes lasts for days) long time hydrodynamic processes are fully absent ? Is it possible that the numerical overestimates of the real data (see Figure 1 (c,d)) is due to this approximation ? I think that the authors need to discuss these issue with some details, certainly in the concluding section but, possibly, also when they describe the numerical results of Figure 1.

Response to Comment (1): We greatly appreciate the reviewer for raising these valuable questions. In the revised manuscript, we have carefully addressed all these constructive comments.

Action for Comment (1):

In Manuscript, Lines 453-461: (in concluding section): “Second, hydrodynamic is reasonably neglected from the modeling (see section “Direct Numerical Simulation”). This reasonable simplifying assumption leads two deviations in the simulation results from experimental observations; (1) simulation predicts that the chiral fronts almost evenly propagate inward while the POM images show that the front propagation is slightly uneven, see Figures 2,3. For this reason, the simulations can overestimate or underestimate the normalized relaxation progress (R) over time evolution, see Figures 1(c,d). (2) The defect movement can't be fully tracked owing to neglecting the weak orientation-induced backflow. However, this insignificantly affects R, or in other words, the relaxation dynamics.”

In Manuscript, Lines 568-579: “Furthermore, the hydrodynamic is reasonably negligible because of three reasons. First, as already explained, the orientational relaxation takes place in the closed system in which; there does not exist any external velocity driving force such as pressure difference, gravity (the filled capillaries were kept horizontally throughout the experiments), moving surface, and so on. Second, as above-mentioned, there is no appreciable concentration gradient in the system; hence, the hydrodynamic induced by mass transportation become negligible. Third, the self-generated transient bulk convection is insignificant owing to following reasons; the viscosity of cholesteric permeation flow is extremely large^{36,77}, the rotational viscosity of BLG and CNC is also considerably large (see Table 2), the essentially vanishing Ericksen number (flow-to-elasticity ratio)⁷⁸, the solid-like behavior along the cholesteric helix axis⁷⁹, and the aposteriory validation and high fidelity of the predictions with experimental data.”

Comment (2): As mentioned by the authors the dynamics of the defects is relevant in establishing the overall relaxation process. However in the paper there is no discussion on this issue. In particular no results are shown on the formation and time evolution of the defects during the relaxation processes of the two systems. To analyse the relaxation process in terms of defects statistics shouldn't be too difficult, at least for the numerical part of the study.

Response to Comment (2): We thank the referee for this comment.

Action for Comment (2):

In Manuscript, Lines 194-210: “Taken altogether, the number of defects is directly and inversely proportional to D/p_{∞} and the relaxation time, respectively. Therefore, it is ideally expected that the defect-less ground state can be achieved as long as D/p_{∞} and the relaxation time are sufficiently small and long, respectively. It should further be taken to account that the defect-less structure is achieved on a defined region of interest ($260\mu\text{m}\times 260\mu\text{m}$) for BLG as shown in Figures 1(a), 2; however, it is impractical to reach the defect-less structure on a large domain (e.g. $600\mu\text{m}\times 260\mu\text{m}$) even after equilibration, see Supplementary Note S2.

Although the defects analysis discussed above was performed on the region of interest ($260\mu\text{m}\times 260\mu\text{m}$), the Supplementary Note S2 demonstrates that the chosen system size is statistically large enough to extend the concluded results regarding the number of trapped defects to larger regions. As can be seen in the Supplementary Figures S2, the number of defects trapped in BLG system is less compared to CNC on a larger region ($600\mu\text{m}\times 260\mu\text{m}$) after equilibration (10 days for BLG and 3 days for CNC). Additionally, the defects existing in the capillary can move and thus can translate in and/or out the system investigated ($260\mu\text{m}\times 260\mu\text{m}$); however, the results already discussed are always statistically valid; see the Supplementary Movies S1 and S2 and the Supplementary Note S2.”

In SI, Lines 48-54:

“**Note S2. Defects Analysis**

collection of the CNC POM images as the system was completely filled with cholesteric birefringence. However, we captured the status of CNC after 3 days. We have addressed this comment (i.e. defect analysis after equilibration) in the previous comment.

Action for Comment (3): Please, see Action for Comment (2)

Comment (4): The results presented here refer to CLC that are confined with strong planar anchoring. What the authors would expect for different anchoring protocols and/or anchoring strength? Can they discuss this point in the discussion section?

Response to Comment (4): Thanks for raising this important comment. We absolutely agree with the referee that BLG and CNC have strong planar anchoring as confirmed by our experimental observations. For this reason, our theory has exhaustively relied on strong anchoring to focus on our experimental observations; hence, we considered a planar boundary condition instead of incorporating surface free energy to the volume free energy. Furthermore, in our view, having precise discussion on the impact of anchoring strength deserves a dedicated study, as previous studies have shown the significance of anchoring strength. However, owing to this valuable comment, we have enriched the manuscript by making the following revisions in regard to the anchoring.

Action for Comment (4):

In Manuscript, Lines 448-453: (in the discussion section): “Before doing so, the following two points should be taken into account regarding the modeling and simulation utilized in this study. First, the results presented here are based on the strong planar anchoring supported by experimental observations. Previous studies⁷⁰⁻⁷⁵ have shown that the effect of anchoring can be significant. Regarding our study, we also anticipate that different anchoring may affect relaxation dynamics; however, it requires a dedicated study.”

In Manuscript, Lines 553-557: “As can readily be appreciated through the POM images (see Figures (1a,b),(2),(3),(4e)), both BLG and CNC have strong planar anchoring with the inner capillary surface. To capture this state, the governing equations are subjected to

$$\mathbf{Q}(\mathbf{x},t)|_{\text{surface}} = \begin{bmatrix} -1/3 & 0 & 0 \\ 0 & 2/3 & 0 \\ 0 & 0 & -1/3 \end{bmatrix} \quad (7)$$

Eq.(7) indicates that fibers located on the surface are aligned parallel to the central axis of the capillary with strong anchoring, $S=1$.”

Comment (5): Minor points: Panel (e) of Figure 4 refer to experimental images or to simulation snapshots. Since the Figure focused on numerical results I suspect that also the images of panel (e) are coming from simulations, but I am not fully sure. Can the authors better clarify this point at least in the figure caption?

Response to Comment (5): We agree with reviewer's comment that adding a clarifying note is advantageous. Following the reviewer's suggestion, we have modified all the figure captions in the revised manuscript.

Action for Comment (5):

In Manuscript, Lines 149-150: "... . The images shown in panels(a,b) were experimentally acquired using the LC PolScope device. ”

In Manuscript, Lines 258-259: "POM images (which are experimentally captured using the LC PolScope device)"

In Manuscript, Lines 281-282: "... . POM images (which are experimentally captured using the LC PolScope device)"

In Manuscript, Line 324: "... . These images were experimentally acquired using the LC PolScope device."

Comment (6): Minor points: The pdf version of the Supplementary Material has some equations whose symbols are not reproduced properly.

Response to Comment (6): We appreciate the referee's comment.

Action for Comment (6): We have carefully read the entire pdf version of manuscript and SI to make sure there is not any inappropriate symbol compilation.

Reviewer #3 (Remarks to the Author):

Referee: This is an important contribution that provides insights into some of the poorly understood aspects of physical behavior of biologically derived cholesteric liquid crystals. The combination of experiments and modeling is a powerful tool to address this behavior. Considering that the results have important insights for many branches of physics, technology, biology and materials science, I believe the manuscript meets the high standards and requirements for NatComm publications in terms of broad interest and novelty. While I think the manuscript should be eventually published, I have some suggestions that I believe are important for authors to consider and account for.

Reply: We greatly appreciate the referee for the positive assessment and constructive suggestions. The manuscript has been carefully revised according to the referee's comments as below. We hope that the revised manuscript would be suitable for publication in Nature Communications.

Comment (1): The title of article might be more general than the scope of work would suggest. For example, cholesteric liquid crystals formed by persistence-length 50nm DNA fragments likely would exhibit different behavior and I see no reason to suggest that the findings described here would be applicable to all biologically derived cholesterics - authors should modify the title to address this.

Response to Comment (1): The referee's comment is correct.

Action for Comment (1): In the revised manuscript, the title has changed to **“Relaxation Dynamics in Bio-Colloidal Cholesteric Liquid Crystals Confined to Cylindrical Geometry”**

Comment (2): Authors use Frank-Oseen-Mermin in the main text and just Frank-Oseen in SI - the reader may be confused and, as a minimum, authors should carefully describe the differences and what they actually.

Response to Comment (2): The explanatory paragraph has been added to the revised Supplementary Information in order to address this legitimate comment.

Action for Comment (2):

In SI, Lines 204-208: “Before proceeding to reveal the curvature role in relaxation dynamics by making use of Frank-Oseen, it should be mentioned that Frank-Oseen is a subset of the \mathbf{Q} -tensor Frank-Oseen-Mermin gradient energy, eq.(6b). These two methods, FO and FOM, can be converted to each other by use of \mathbf{Q} -tensor definition, eq.(S5), and mapping between L_i and K_i discussed in references^{10,11}”

Comment (3): The literature review and discussion seem to be missing the discussion of potential technological implications of this work. For example, control of equilibrium and metastable states in the forms of cholesterol and paranematic structures is important for forming retarder films and cholesterol color filters [ACS PHOTONICS 5, 2468-2477 (2018)] as well as for plasmonic mesostructured materials [ADVANCED MATERIALS 26, 7178-7184 (2014)].

REVIEWERS' COMMENTS:

Reviewer #1 (Remarks to the Author):

The authors thoroughly answered my comments.
In my opinion, the manuscript is now better and more complete.
This is a beautiful combination of experiments and numerical simulations.

I think that my comment (13) was misunderstood, but it is not a major issue.
The point was not to have the authors add a discussion about the inevitable presence of defects in the samples (the discussion is fine).
Perhaps I should rephrase it:
Starting from a fully PN state, twisting by one half-pitch seems topologically impossible by just rotating the director without creating a singularity. To do so, the systems has two options, both of which are apparently observed:
1) either it creates singularities, which means it destroys the orientational order locally (Isotropic phase). This corresponds to "melting" the liquid crystal and is why I wrote the comment about confinement-induced phase transitions. Since the order parameter usually vanishes inside a disclination, the isotropic phase in the middle of the tube could perhaps be called a giant disclination.
2) or it propagates defects already present in the sample or generated somewhere, similar to the propagation of Grandjean-Cano lines forming a loop. This corresponds to having fronts propagating and results in a "slow-fast" mechanism.

About the answer to comment (1):
In the first case 1), the N^* to I transition is induced by an excessive elastic energy, as in confinement-induced phase transitions. Confined systems are not submitted to an external stress, but they are submitted to an extra internal stress because they have to comply with the boundary conditions.
In the second case 2), although the PN state is less favorable than the N^* state, I do not think it can be called "unstable" (no nucleation is observed). I believe it is metastable instead. The propagation of pi-walls between states of unequal twist is a well known phenomenon (for instance Lam, L. Chaos, Solitons & Fractals, 5 (1995), 2463-2473.).
This is why I am still not fully convinced that the observations are as unexpected as the authors claim. This is arguable.

In the end, my recommendation is the following: This is an interesting, serious, complete and beautiful piece of work. I have no suggestions about how to improve the manuscript. I do not see it however as revolutionary.

Minor comments:

"aposteriory" should be "a posteriori".

Shouldn't "the hydrodynamic" be "the hydrodynamics" (twice on page 23)?

Reviewer #2 (Remarks to the Author):

After having carefully read the present manuscript, in its revised form, as well as the correspondence between editors, referees and authors, I think that the authors have addressed many of my previous concerns and greatly improved the manuscript.

Hence I suggest its publication in its present form.

Reviewer #3 (Remarks to the Author):

authors accounted for my suggestions and I recommend publication as is

Authors' Response to the Referees' Reports

other hand, for CNC the order parameter is nearly zero ($S=0.01-0.05$) and the PN state is essentially isotropic with very weak anisotropy. This fact is at the core of the phenomena we measured and explained.

Comment (2): About the answer to comment (1):

In the first case 1), the N^* to I transition is induced by an excessive elastic energy, as in confinement-induced phase transitions. Confined systems are not submitted to an external stress, but they are submitted to an extra internal stress because they have to comply with the boundary conditions.

In the second case 2), although the PN state is less favorable than the N^* state, I do not think it can be called "unstable" (no nucleation is observed). I believe it is metastable instead. The propagation of π -walls between states of unequal twist is a well known phenomenon (for instance Lam, L. Chaos, Solitons & Fractals, 5 (1995), 2463-2473.).

This is why I am still not fully convinced that the observations are as unexpected as the authors claim.

This is arguable.

Response to Comment (2):

Regarding "In the first case 1), the N^* to I transition is induced by an excessive elastic energy, as in confinement-induced phase transitions.", we would like to summarize all the processes which take place in our study according to the timeline for BLG and CNC.

A. BLG Material. (1) Before the relaxation dynamic starts, for $t<0$, corresponding to the capillary filling stage: the N^* phase, which is in the bottle shown in Supplementary Figure 16, is sucked into the capillary tube by the capillary force. During the suction, the well-known flow-induced N^* -PN transition takes place. Hence, at the beginning of relaxation, $t=0$, there are two phases in the capillary tube: first, N^* near the wall, and the PN in the center. (2) As the referee mentioned, owing to the excessive elastic energy in the PN phase, the PN phase lowers the order parameter to decrease the excessive elastic energy. We denote the low order PN as LOPN for simplicity. It should be noted that the excess energy in our study is due to the intrinsic chirality of the ground state, meaning that $p_\infty \rightarrow \infty$ for PN whereas $p_\infty = \text{finite}$ at the ground state. (3) Finally, the LOPN- N^* transition takes place through a moving front.

In partial summary, for BLG there are three sequential transformations in our study as follows: N^* -PN, PN-LOPN, and LOPN- N^* . The first two are due to the capillary flow and the

excess elastic energy relief, respectively. The latter is spontaneous.

B. CNC Material: The phenomena involved are now different in CNC than in BLG because the initial PN phase is transformed into a nearly isotropic phase with an order parameter of the order of 0.01-0.05, and hence the anisotropy level is weak. We again identify three stages. 1) Before the relaxation dynamic starts, for $t < 0$, corresponding to the capillary filling stage: the N^* phase, which is in the bottle shown in Supplementary Figure 16, is sucked into the capillary tube by the capillary force. During the suction, the well-known flow-induced N^* -PN transition takes place. Hence, at the beginning of relaxation, $t=0$, there are two phases in the capillary tube: first, N^* near the wall, and the PN in the center. (2) As the referee mentioned, owing to the excessive elastic energy in the PN phase, the PN phase lowers the order parameter to decrease the excessive elastic energy. The order parameter is of the order of 0.01-0.05 and the phase is essentially isotropic, I, with essentially no chirality, and no anisotropy. (3) Finally, the I- N^* transition takes place through a monotonic moving front.

In partial summary, for CNC there are three sequential transformations in our study as follows: N^* -PN, PN-I, and I- N^* . The first two are due to the capillary flow and the excess elastic energy relief, respectively. The latter is spontaneous.

For the above-mentioned reasons, we do not ascribe the transformations existing in our study to confinement; however, as we have explicitly elaborated in our manuscript and Supplementary Note 10, confinement has a profound impact on the relaxation dynamics.

Regarding “Confined systems are not submitted to an external stress, but they are submitted to an extra internal stress because they have to comply with the boundary conditions.”, we agree with the referee regarding the existence of internal stress. In Supplementary Note 15, we have mathematically proved that the internal stress is embedded in our modeling, **Q**-tensor equations.

Regarding “In the second case 2), although the PN state is less favorable than the \$N^*\$ state, I do not think it can be called "unstable" (no nucleation is observed). I believe it is metastable instead.”, the PN phase is unstable. Our reason is as follows: the PN phase is an unwound

Reviewer #3 (Remarks to the Author), Second Revision:

Referee: authors accounted for my suggestions and I recommend publication as is

Reply: We would like to thank the Reviewer very much for his/her careful reading of our manuscript and support of our work.